# Genomic Selection for End-Use Quality and Processing Traits in Soft White Winter Wheat Breeding Program with Machine and Deep Learning Models

**DOI:** 10.3390/biology10070689

**Published:** 2021-07-20

**Authors:** Karansher Singh Sandhu, Meriem Aoun, Craig F. Morris, Arron H. Carter

**Affiliations:** 1Department of Crop and Soil Sciences, Washington State University, Pullman, WA 99164, USA; k.sandhu@wsu.edu (K.S.S.); meriem.aoun@wsu.edu (M.A.); 2USDA-ARS Western Wheat Quality Laboratory, E-202 Food Quality Building, Washington State University, Pullman, WA 99164, USA; morrisc@wsu.edu

**Keywords:** deep learning, end-use quality, genomic selection, machine learning, wheat breeding

## Abstract

**Simple Summary:**

Wheat (*Triticum aestivum* L.) breeding programs mainly focus on improving grain yield, biotic and abiotic stress tolerance, and end-use quality traits. End-use quality and processing traits are the combinations of various predefined parameters. Genomic selection (GS) opens the potential for selecting improved end-use quality lines. This study explored the potential of the machine and deep learning-based GS models for predicting end-use quality traits in wheat. Deep models were superior to traditional statistical and Bayesian models under all the prediction scenarios. The high accuracy observed for end-use quality traits in this study support predicting them in early generations, leading to the advancement of superior genotypes to more extensive grain yield trails.

**Abstract:**

Breeding for grain yield, biotic and abiotic stress resistance, and end-use quality are important goals of wheat breeding programs. Screening for end-use quality traits is usually secondary to grain yield due to high labor needs, cost of testing, and large seed requirements for phenotyping. Genomic selection provides an alternative to predict performance using genome-wide markers under forward and across location predictions, where a previous year’s dataset can be used to build the models. Due to large datasets in breeding programs, we explored the potential of the machine and deep learning models to predict fourteen end-use quality traits in a winter wheat breeding program. The population used consisted of 666 wheat genotypes screened for five years (2015–19) at two locations (Pullman and Lind, WA, USA). Nine different models, including two machine learning (random forest and support vector machine) and two deep learning models (convolutional neural network and multilayer perceptron) were explored for cross-validation, forward, and across locations predictions. The prediction accuracies for different traits varied from 0.45–0.81, 0.29–0.55, and 0.27–0.50 under cross-validation, forward, and across location predictions. In general, forward prediction accuracies kept increasing over time due to increments in training data size and was more evident for machine and deep learning models. Deep learning models were superior over the traditional ridge regression best linear unbiased prediction (RRBLUP) and Bayesian models under all prediction scenarios. The high accuracy observed for end-use quality traits in this study support predicting them in early generations, leading to the advancement of superior genotypes to more extensive grain yield trails. Furthermore, the superior performance of machine and deep learning models strengthens the idea to include them in large scale breeding programs for predicting complex traits.

## 1. Introduction

Wheat (*Triticum aestivum* L.) breeding programs mainly focus on improving grain yield, biotic and abiotic stress tolerance, and end-use quality traits. Hexaploid wheat is classified into hard and soft wheat classes based on kernel texture, milling quality, protein strength, and water absorption [1,2,3]. Washington State was ranked fourth in U.S. wheat production in 2020. About 80% of wheat grown in eastern Washington is soft white wheat (SWW), one of the six classes grown in the USA. SWW is the smallest wheat class and is consistently in demand from overseas markets owing to its end-use quality attributes. More than 85% of the SWW produced in the Pacific Northwest (PNW) region is exported to markets in countries like Japan, Korea, The Philippines, and Indonesia [4].

End-use quality and processing traits are the combinations of various predefined parameters [5]. Milling traits are measured to extract flour and break flour percentage as flour yield and break flour yield [6]. Thermogravimetric ovens are used for measuring the flour ash. Lower flour ash is recommended as higher ash levels are indicative of higher bran contamination, which reduces the functionality of most dough and batters [6]. The sugar snap cookie test is an essential test for evaluating SWW to meet expectations of product performance from overseas markets. Baking of cookies is performed for lines within the breeding program, and SWW lines having cookie diameter above 9.3 cm is preferred [7].

Grain characteristics commonly measured in SWW include kernel hardness, kernel size, kernel weight, test weight, and grain protein content. Kernel weight, kernel size, and kernel texture (hardness) are measured with a single kernel characterization system (SKCS). Lower values from the SKCS demonstrate softness; thus, SKCS values are negatively correlated with break flour yield. However, the two measures of kernel texture are not entirely correlated because SKCS includes only kernel resistance to crushing, whereas break flour yield includes particle size, sieving, and grain structure [8]. High gluten strength or viscoelastic strength is required for bread baking, whereas confectionary products require less gluten and water absorption. Gluten strength and water absorption capacity are measured using sodium dodecyl sulfate sedimentation and water solvent retention capacity tests. Lower water absorption in SWW contributes to better cookie spread [2,5].

Major genes influencing end-use quality traits are typically already fixed in most breeding programs, especially in different market classes. Until now, marker-assisted selection has been used for major genes controlling end-use quality, namely, low molecular weight glutenins, high molecular weight glutenins, granule bound starch synthase I (amylose composition), and puroindolines (kernel hardness) [2,9]. Usage of these molecular markers only aid in differentiating different wheat classes earlier in the breeding program; however, they do not provide the complete profile of different end-use quality traits. Previous linkage mapping and genome-wide association studies in SWW have shown that a large number of small effect QTLs control most end-use quality traits in addition to the already fixed genes [4,10,11]. Similarly, 299 small effect QTLs were identified using multi-locus genome-wide association studies for nine end-use quality traits in hard wheat [12]. Kristensen et al. [13] were unable to identify significant QTLs for Zeleny sedimentation, grain protein content, test weight, thousand kernel weight, and falling number in wheat and suggested genomic selection as the best alternative for predicting quantitative traits.

Genomic selection (GS) opens up the potential for selecting improved end-use quality lines due to the small effect of these loci, limited seed availability earlier in the breeding pipeline for conducting tests, and time constraints in winter wheat for sowing the new cycle [14]. GS uses the genotypic and phenotypic data from previous breeding lines or populations to train predictive statistical models. These trained models are subsequently used to predict the genomic estimated breeding values (GEBVs) of genotyped lines [15]. GS has shown the potential to enhance genetic gain by reducing the breeding cycle time and improving selection accuracy [16,17,18]. This is especially important for winter wheat end-use quality traits, as phenotyping requires more than three months and data from the quality lab is often not available between harvest and planting. This ultimately results in either the increase of one year in the breeding cycle or passage of undesirable lines into the next growing season. Furthermore, phenotyping requires a large amount of seed and is costly, so large-scale testing is often not conducted until later generations. Currently, the cost of genotyping 10,000 lines with high density genotyping by sequencing is equivalent to phenotyping 200 lines for end-use quality and processing traits [5]. Therefore, GS is currently the best option for breeding for end-use quality traits after considering time, cost, and seed amount.

Genomic selection has been primarily explored in several hard wheat end-use quality trait studies using the traditional genomic best linear biased prediction (GBLUP), Bayes A, Bayes B, Bayes C, and Bayes Cpi, showing mixed results, where one model performed best for one trait and not for another [19,20]. Machine and deep learning models have opened up an entirely new platform for plant breeders and exploring them in the breeding program could accelerate the pace of genetic gain. Deep learning models have shown higher prediction accuracies for different complex traits in wheat [21], rice (*Oryzae sativa* L.) [22], soybean (*Glycine max* L.) [23], and maize (*Zea mays* L.) [24]. Sandhu et al. [21] have shown that two deep learning models, namely, convolutional neural network (CNN) and multilayer perceptron (MLP), gave 1–5% higher prediction accuracy compared to BLUP-based models. Ma et al. [25] and Montesinos-López et al. [26,27] also obtained similar results to predict quantitative traits in wheat and suggested that deep learning models should be explored due to their better prediction accuracies. To the best of our knowledge from our literature search, this is the first study exploring the potential of the deep learning models for predicting the end-use quality traits in wheat.

This study explored the potential of GS using multi-environment data from 2015-19 for end-use quality traits in a soft white winter wheat breeding program. We explored nine different BLUP based models, Bayesian models, and machine and deep learning models to predict the fourteen different end-use quality traits. The main objectives of this include, (1) Optimization of the machine and deep learning models for predicting end-use quality traits, (2) Comparison of prediction ability of nine different GS models to predict fourteen different end-use quality traits using cross-validation approaches, and (3) Assess the potential of GS for forward prediction and across location predictions using previous years training data in the breeding program.

## 2. Materials and Methods

### 2.1. Germplasm

A total of 666 soft white winter wheat lines were evaluated for five years at two locations, namely, Pullman and Lind, WA, USA, from 2015–19. These 666 genotypes consist of F_4:5_ lines, double haploid lines, lines in preliminary and advanced yield trials screened as a part of the Washington State University winter wheat breeding program. F_4:5_ derived lines and double haploid lines were screened for the agronomic and disease resistance traits, and the superior genotypes were tested for the end-use quality. Lines in preliminary and advanced yield trials were selected for superior yield, and those lines were later advanced for end-use quality trait phenotyping. Some genotypes were replicated at a single location per year, whereas others were un-replicated, creating an unbalanced dataset [28]. As this was a breeding population, very few lines were common between all the years except check cultivars due to continuous selections.

### 2.2. Phenotyping

Fourteen different end-use quality and processing traits were measured, and data were obtained from the USDA-ARS Western Wheat & Pulse Quality Laboratory, Pullman, WA. All these traits were measured following the guidelines of the American Association of Cereal Chemists International (AACCI 2008). These fourteen traits were divided into four categories: milling traits, baking parameters, grain characteristics, and flour parameters. The complete summary of each trait, number of observations, mean, standard error, and heritability is provided in Table 1 and Table 2.

Grain characteristics, namely kernel size (KSIZE), kernel weight (KWT), and kernel hardness (KHRD) were determined using 200 seeds/sample with a SKCS 4100 (Perten Instruments, Springfield, IL, USA) (AACC Approved Method 55-31.01). Grain protein content (GPC) was measured using a NIR analyzer (Perten Instruement, Springfield, IL, USA) (AACC Approved Method 39-10.01). Test weight (TWT) was obtained as weight/volume following AACC Approved Method 55-10.01.

Three milling traits, namely flour yield (FYELD), break flour yield (BKYELD), and milling score (MSCOR) were obtained using a Quadrumat senior experimental mill (Brabender, South Hackensack, NJ, USA). FYELD was determined as a ratio of total flour weight (mids + break flour) to the initial sample weight using a single pass through the Quadrumat break roll unit. BKYELD was estimated as the percent of milled product passing through a 94-mesh* screen per unit grain weight. Flour ash (FASH) was obtained using the AACC Approved Method 08-01.01. MSCOR was calculated using the formula: MSCOR = (100 − (0.5(16 − 13.0 + (80 − FYELD) + 50 (FASH − 0.30))) × 1.274) − 21.602, showing that this trait is a function of FYELD and FASH content. To evaluate baking parameters, cookie diameter (CODI) was measured using AACC Approved Method 10-52.02.

Four different flour parameters, namely, flour protein (FPROT), water solvent retention capacity in water (FSRW), flour swelling volume (FSV), and flour sodium dodecyl sulfate sedimentation (FSDS) were measured from the extracted flour. FPROT was measured following the AACC Approved Method 39-11.01. FSRW measures the water retention capacity of gluten, gliadins, starch, and arabinoxylans using the AACC Approved Method 56-11.02. The FSDS test was used to measure strength of gluten by following the AACC Approved Method 56-60.01. The FSV test assesses starch composition following the AACC Approved Method 56-21.01 [29].

### 2.3. Statistical Analysis

Due to the unbalanced nature of the dataset, adjusted means were calculated using residuals obtained using the lme4 R package for within environment analysis. The model equation is represented as
Y_ij_ = Block_i_ + Check_j_ + e_ij_(1)
where Y_ij_ is the raw phenotype; Check_j_ is the effect of replicated check cultivar; Block_i_ corresponds to the fixed block effect; and e_ij_ is the residuals [30,31]. Block was considered fixed, as we want to remove that component of variation before exploring the genetic variation. Residuals from the model were used to calculate the adjusted means (line effect).

Adjusted means across the environments were calculated following the method implemented in Sandhu et al. [18,32] and is as follows
Y_ijk_ = µ + Block_i_ + Check_j_ + Env_k_ + Block_i_ × Env_k_ + Check_j_ × Env_k_ + e_ijK_(2)
where Y_ijk_ is the raw phenotype value; Block_i_, Check_j_, and Env_k_ are the fixed effect of ith block, jth check, and kth environment; and e_ijk_ is the residuals. Residuals from the model were used to calculate the adjusted means (line effect).

Best linear unbiased predictors (BLUPs) for individuals and across environments were used to obtain the variance components for estimating broad sense heritability. The equation for heritability used was
(3)$$H_{C}^{2}=1-\frac{{\bar{v}}_{∆..}^{\mathrm{BLUP}}}{2\sigma_{g}^{ˆ2}}$$
where $H_{C}^{2}$ is the Cullis heritability; σg^2  is genotypic variance; and ${\bar{v}}_{∆..}^{\mathrm{BLUP}}$ is the mean-variance of BLUPs [33].

### 2.4. Genotyping

The whole population was genotyped using GBS through the North Carolina State University (NCSU) Genomics Sciences Laboratory, Raleigh, NC, using the restriction enzymes *Pst*I and *Msp*I [34]. LGC Biosearch Technologies Oktopure^TM^ robotic platform with sbeadex^TM^ magnetic microparticle reagent kits were used to extract the DNA from the leaves of ten-day-old seedlings. Thermo Fisher (Waltham, MA, USA) Quant-It^TM^ PicoGreen^TM^ assays were used to quantify the DNA, and the samples were normalized to 20 ng/µL. Restriction enzymes *Pst*I and *Msp*I were used for sample fragmentation, and the digested samples were ligated with barcode adapters using T4 ligase. The pooled samples were amplified using PCR, following Poland et al. [34], and sequencing was performed at NCSU Genomics Sciences Laboratory. Burrows-Wheeler Aligner (BWA) 0.7.17 was used to align the sequences to the Chinese Spring (IWGSC) RefSeq v1.0 reference genome [35]. Tassel v5 was used for SNP discovery and calling [36]. Quality filtering pipeline was implemented in R software to remove markers with minor allele frequency less than 5%, markers missing more than 30% data, and heterozygosity more than 15%. Missing data in the SNP data were imputed using the expectation-maximization algorithm in the R package rrBLUP [37]. After the complete filtering pipeline, 40,518 SNPs remained and were used for population structure and genomic prediction [28].

### 2.5. Genomic Selection Models

We explored the performance of five parametric and four non-parametric models for all fourteen traits evaluated in this study. Parametric models used were RRBLUP, Bayes B, Bayes A, Bayes Lasso, and Bayes C. Non-parametric models included two machine and two deep learning models. The complete information for all those models and optimization process is provided as follows:

#### 2.5.1. Ridge Regression Best Linear Unbiased Prediction (RRBLUP)

RRBLUP was included here as the benchmark for comparing its performance with other models due to frequent use in wheat breeding and ease of implementation. The model assumes that all markers contribute to the trait and has a constant effect variance. Marker effects and variance patterns are estimated using the restricted estimated maximum likelihood (REML) function based on phenotypic and marker data [37]. The RRBLUP model was implemented with the R package rrBLUP using the *mixed.solve* function. The model can be represented as
(4)y=µ+Zu+e
where µ is the overall mean; y is the vector of adjusted means; u is a vector with normally distributed random marker effects with constant variance as u ~ N(0, Iσ^2^_u_); Z is an N × M matrix of markers; and e is the residual error distributed as e ~ N(0, Iσ^2^_e_). The solution for mixed equation can be written as
(5)$$u=Z^{T}\;{\left({\mathrm{ZZ}}^{T}\;+\;\lambda I\right)}^{-1}\;y$$
where u, Z, and y are explained above; I is an identity matrix; and λ is represented as λ = σ^2^_e_/σ^2^_u_ and is the ridge regression parameter [37].

#### 2.5.2. Bayesian Models

We implemented four different Bayesian models, namely, Bayes Lasso, Bayes A, Bayes B, and Bayes C. All these models assume different prior distributions for estimating marker effects and variances. Bayes A applies the inverted chi-squared probability distribution for estimating marker variances. Bayes B provides a more realistic scenario for breeding, assuming that all markers do not contribute to the phenotype. It applies a mixture of prior distribution with a high probability mass at zero, and others follow the Gaussian distribution. Bayes C and Bayes Lasso follow the mixture of the prior distribution (point mass at zero with scaled-t distribution) and double exponential distribution, respectively [38]. All the Bayesian models were implemented using the BGLR R package using the model equation
(6)$$y_{i}=µ+\overset{j=p}{\underset{j=1}{\sum }}x_{ij}\beta_{j}+\epsilon_{i}$$
where µ, $y_{i}$, $x_{ij},$ and $\epsilon_{i}$ are defined above; and $\beta_{j}$ is the jth marker effect. Each Bayesian model used in this study has separate conditional prior distribution. Analysis was performed with 30,000 Monte Carlo Markov chain iterations with 10,000 burn-in iterations [38].

#### 2.5.3. Random Forests (RF)

RF involves building a large collection of identical distributed trees and averages from the trees for final prediction. Different bootstrap samples are performed over the training set to identify the best feature subsets for splitting the tree nodes. The main criteria for splitting at the node include lowering the loss function during each bootstrapped sample [39]. Model equation is represented as
(7)$${\hat{y}}_{i}=\frac{1}{B}\overset{B}{\underset{b=1}{\sum }}T_{b}\left(x_{i}\right)$$
where ${\hat{y}}_{i}$ is the predicted value of the individual with genotype $x_{i}$; *T* is the total number of trees; and *B* is the number of bootstrap samples. The main steps involved in model functioning includes

Bootstrap sampling (b = (1, …, B)) to select genotypes with replacement from the training set, and an individual plant can appear once or several time during the samplingBest set of features (SNP_j_, j = (1, …, J) were selected to minimize the mean square error (MSE) using the max feature function in the random forest regression library.Splitting is performed at each node of the tree using the SNP_j_ genotype to lower the MSE.The above steps are repeated until a maximum depth is reached or a minimum node. The final predicted value of an individual of genotype $x_{i}$ is the average of the values from the set of trees in the forest.

The important hyperparameter model training includes the depth of the trees, the importance of each feature, the number of features sampled for each iteration, and the number of trees. Randomized grid search cross-validation was used for hyperparameter optimization. The combination of hyperparameters that were tried included max depth (40, 60, 80, 100), max features (auto, sqrt), and number of trees (200, 300, 500, 1000) [40]. The Scikit learn, and random forest regression libraries in Python 3.7 were used for analysis [41].

#### 2.5.4. Support Vector Machine (SVM)

SVM uses the non-linear kernel for mapping the predictor space to high dimensional feature space for studying the relationship between marker genotype and phenotypes. The model equation is represented as
(8)f(x)=wx+b
where f(x) is learning function; *b* is the constant, reflecting the maximum allowed bias; *w* is the unknown weight; and x is the marker set. The learning function is mapped by minimizing the loss function as
(9)$$C\overset{n}{\underset{i=1}{\sum }}L\left(e_{i}\right)+\frac{1}{2}{\left\|w\right\|}^{2}$$
where *C* is a positive regularization parameter; ${\left\|w\right\|}^{2}$ represents model complexity, $e_{i}$ = y − f(x) is the associated error with the *i*th training data point, and L  is the loss function [42].

#### 2.5.5. Multilayer Perceptron (MLP)

MLP is the feed-forward deep learning model that uses three layers, namely, input, hidden, and output, for mapping the relationship. These layers are connected by a dense network of neurons, where each neuron has its characteristic weight. MLP uses the combination of neurons, activation function, learning rate, hidden layers, and regularization for predicting the phenotypes. Input layer corresponds to SNP genotypes, while neurons connect multiple hidden layer with associated strength (weight). The output of the *i*th hidden layer is represented as
(10)$$Z_{i}=b_{\left(I-1\right)}+W_{i}\;f_{\left(I-1\right)}\;(x)$$
where *Z_i_* is the output from the *i*th hidden layer; b_0_ is the bias for estimating neurons weight; *f*_(*I*−1)_ represents the activation function; and *W_i_* is the weight associated with the neurons, and this process is repeated until the output layer.

Keras function’s grid search cross-validation and internal capabilities were used for optimizing the hyperparameters. Hyperparameters giving the lowest MSE were identified and used for output prediction [43]. Regularization, dropout, and early stopping were applied to control overfitting. Furthermore, information about hyperparameter optimization and deep learning models is referred to in [21,31].

#### 2.5.6. Convolutional Neural Network (CNN)

CNN is a special case of a deep learning model that accounts for the specific pattern present between the input features. Information about the CNN model, its implementation, and hyperparameter optimization are referred to in previous publications [21,31]. A combination of input, convolutional, pooling, dense, flatten, dropout, and output layers were applied for the prediction. Like MLP, hyperparameter was optimized using grid search cross-validation to select filters, activation function, solver, batch size, and learning rate. Regularization, dropout, and early stopping were applied to control overfitting. All the deep learning algorithms were implemented using Scikit learn and Keras libraries [44,45].

### 2.6. Prediction Accuracy and Cross-Validation Scheme

Prediction accuracy was evaluated using five-fold cross-validation where 20% of the data were used for testing and the remaining 80% for training within each environment. One hundred replications were performed for assessing each model’s performance. One replicate consisted of five iterations where data are split into five different groups. Prediction accuracy was reported as the Pearson correlation coefficient between the true (observed phenotype) and GEBVs. Separate analysis was performed for both locations using a cross-validation approach to assess the model’s performance.

Independent predictions or forward predictions were performed by training the model on previous year data and predicting future environments (i.e., 2015 data from Lind was used to predict 2016; 2015 and 2016 data predict 2017, and so on for both locations). In the end, we tried to predict the 2019 environment of both locations by using the whole data set from the other location (i.e., 2015–19 data from Lind was used to predict 2019 in Pullman). Forward prediction represents real prediction scenarios in breeding programs where previous data are used to predict future environments. Due to computational burden, all the GS models were analyzed over the Kamiak high-performance cluster (https://hpc.wsu.edu/, accessed on 10 April 2021).

## 3. Results

### 3.1. Phenotypic Data Summary

Table 1 provides information on the number of lines screened for end-use quality traits across years at Lind and Pullman. One thousand three hundred thirty-five lines were phenotypically screened for end-use quality traits across five years (2015–19) at two locations (Table 1). Overall, Pullman had more lines compared to Lind for each year. Summary statistics, including mean, minimum, maximum, standard error, and heritability are provided for all the fourteen end-use quality traits (Table 2). Broad sense heritability ranged from 0.56 to 0.93 for different traits. All the traits were highly heritable except GPC and FPROT (Table 2).

Significant positive and negative correlations were observed among different traits (Figure 1). Positive correlations were observed between FYELD and BKFYELD, KSIZE and KWT, GPC and FPROT, FSDS and FPROT, GPC and FSDS, and FSRW and KHRD (Figure 1). Similarly, negative correlations were seen between FASH and MSCOR, CODI and KHRD, GPC and FSV, FSDS and CODI, and CODI and FSRW (Figure 1). Most of the traits were not strongly correlated with each other, suggesting that a single quality trait cannot substitute for others; hence, measurements from all of them are required for selection decisions.

### 3.2. Cross-Validation Genomic Selection Accuracy and Model Comparison

Complete datasets across the years from Pullman and Lind were used to predict the fourteen end-use quality traits using nine different models (Table 3, Figure 2). Five-fold cross-validation was performed to compare the results from the models at both locations. Prediction accuracy at Pullman varied from 0.52–0.81 for all traits with nine different GS models. The highest prediction accuracy was 0.81 for KWT and KSIZE with the RF and MLP model at Pullman (Figure 2). The lowest prediction accuracies were for GPC, FASH, FPROT, and FSRW at Pullman using different GS models (Table 3). The highest prediction accuracy for each trait is bolded for comparison with other models (Table 3). For the fourteen end-use quality traits evaluated in this study at Pullman, deep learning models, namely MLP and CNN, performed best for eight of the traits, demonstrating the potential to incorporate them into breeding programs (Table 3) for prediction. RF and SVM performed best for three and four traits out of the fourteen, respectively, while RRBLUP performed superior for only one trait (Table 3 and Figure 2).

Prediction accuracies (0.45–0.70) within the Lind dataset were lower than Pullman for all traits (Table 2). Similar to Pullman, the highest cross-validation prediction accuracy (i.e., 0.70) was obtained for KWT at Lind. The lowest prediction accuracies were obtained for GPC, FPROT, and FSRW using the Bayesian models (Table 3). Machine and deep learning models performed superior for twelve out of the fourteen end-use quality traits (Figure 2). Table 3 provides the average performance for all models, and we observed that machine and deep learning models performed superior to RRBLUP and all the Bayesian models. On average, machine and deep learning models performed 10 and 5%, superior to Bayesian and RRBLUP.

### 3.3. Forward Predictions

GS model predictions were assessed to reflect the power of training size to predict the phenotypes in future years. Figure 3 and Figure 4 show the results for forward predictions at Pullman and Lind when combined data from the previous years were used to predict the phenotypes. The *x*-axis represents the year for which predictions were made while training the models on all the previous year’s phenotypic data (Figure 3 and Figure 4; Appendix A). We saw a gradual increase in prediction accuracy for all the traits as we kept increasing the training data size, and the same trend was observed for both locations (Figure 3 and Figure 4). The highest improvement in prediction accuracy was observed for GPC, FPROT, FASH, and FSDS, owing to the complex nature of these traits and demonstrating the importance of training size. Similar to cross-validation prediction accuracy (Table 3), the highest forward prediction accuracy was obtained with machine and deep learning models, especially when the training data size kept increasing (Figure 3 and Figure 4). Bayesian models performed worst for all of the traits and at both locations, even when training data size was increased.

Forward predictions in 2019 were, on average, 32 and 29% greater than forward predictions in 2016 for Pullman and Lind (Figure 3 and Figure 4). The highest improvement in forward predictions from 2016 to 2019 was 0.35 to 0.55 for CODI, while the lowest was 0.26 to 0.29 for KWT (Figure 3). The highest improvement was seen for MLP and CNN, demonstrating as the size of training data increases, deep learning models result in the highest improvement in prediction accuracy. Furthermore, cross-validation prediction accuracies were, on average, 34 and 32% higher than forward prediction in 2019 for Pullman and Lind (Table 3, Figure 3 and Figure 4), suggesting that cross-validation scenarios over-inflate prediction accuracies.

### 3.4. Across Location Predictions

Across location predictions were performed where data from Lind was used to train the model for predicting performances in Pullman and vice versa. Owing to all the Bayesian model’s worst performance and computational burden in cross-validation and forward predictions, these models were eliminated for the across location predictions. Figure 5 and Table 4 showed the prediction accuracy for all fourteen end-use quality traits when predictions were made for 2019_Pullman by models training on the whole Lind dataset and vice versa. The across location prediction accuracies were, on average, 16 and 47% less than forward and cross-validation prediction accuracies, demonstrating the importance of inclusion of genotype by environment interaction components into the GS models for across location and environment predictions.

Deep learning models performed best for across location prediction compared to RRBLUP and machine learning models (Table 4 and Figure 5). The highest prediction accuracy was 0.50 for FYELD with a MLP model for predicting 2019_Pullman (Table 4). The lowest prediction accuracies were for MSCOR, GPC, and FSV with the RRBLUP model for predicting 2019_Pullman (Table 4). Out of the four models used, twelve end-use quality traits were best predicted by deep learning models under the 2019_Pullman scenario, while RF performed best for the remaining two traits (Table 4). In 2019_Lind predictions, the highest accuracy was 0.50 for FYELD with the MLP model, and the lowest was for GPC and MSCOR with the RRBLUP model. Similar to 2019_Pullman, deep learning models performed best for eleven out of the fourteen traits evaluated in 2019_Lind.

## 4. Discussion

Selection for end-use quality traits is often more difficult to conduct compared to grain yield, disease resistance, and agronomic performance, due to the cost, labor, and seed quantity requirements [46]. Phenotyping for quality traits is usually delayed until later generations, resulting in creating small population sizes with unbalanced datasets [16]. This study explored the potential of GS, especially machine and deep learning models, for predicting fourteen different end-use quality traits using five years (2015–19) of phenotyping data from a winter wheat breeding program. The prediction accuracy in this study varied from 0.27–0.81, demonstrating the potential of its implementation in the breeding program. We observed that forward and across location prediction accuracies could be increased using deep and machine learning models without accounting for genotype by environment interaction, environment covariates, and kernel matrices in traditional GS models. Furthermore, QTLs or major genes controlling quality traits are typically already fixed in the particular market class or breeding programs; hence, GS is the best substitute for marker-assisted selection by exploring different combinations of QTL to achieve the best variety [47].

The broad-sense heritability of end-use quality traits evaluated varied from 0.56 to 0.93, with the majority of them having a value above 0.80. Similar heritability values were obtained by Michel et al. [48], Jernigan et al. [49], and Kristensen et al. [50] for different baking and flour yield parameters of winter wheat. These intermediate to high heritability estimates suggested that most of the variation in these traits is attributed to genetic factors and considerably less affected by environment and genotype by environment interactions [51]. Therefore, GS is the best option for predicting these traits due to capturing most of the additive genetic variation by the models, as observed in this study, due to intermediate to high prediction accuracy for different quality traits. We observed that only a few grain and flour assessments traits were correlated. These low correlations among most end-use quality traits strengthen the fact that no single quality parameter can assist in final variety selection, but that many are needed [1]. Only three end-use quality traits, namely, GPC, FPROT, and FSV, had intermediate heritability values, which were also reported in previous studies due to their complex and polygenic inheritance nature [32,52]. Similarly, comparatively low prediction accuracies obtained from these traits validated the fact for inclusion of genotype by environment interaction or environmental covariates for their prediction [53].

Cross-validation prediction accuracies were, on average, 34% and 32% higher than forward prediction in 2019 for Pullman and Lind. The higher cross-validation prediction accuracies compared to forward and across location prediction suggests the importance of including bigger training sets, genotype by environment interactions, and environment covariates for exploiting the maximum variation to make predictions [54]. Higher accuracies obtained in cross-validation showed that most of those values are over-inflated, and attention is required before making any final decision about those large values to adopt GS in the breeding program [55]. Cross-validation approaches included training and testing sets from the same environment, thus accounted for environmental variation in prediction. Moreover, most of the lines evaluated in breeding programs are usually closely related or full sibs and confound cross-validation approaches, where full sibs might be in the same training or testing group, causing inflation in prediction accuracies [56]. The relationship between individuals in the training and testing set profoundly affects model performance, with a closer relationship resulting in higher accuracy. Forward and across location prediction are the best methods for studying the importance of GS implementation in the breeding program [57,58].

Continuous increments in forward prediction accuracy with all nine models demonstrated the importance of a large training population and more environments for training the GS model [59]. He et al. [60] and Battenfield et al. [16] observed an increase in forward prediction in spring wheat end-use quality traits. Similarly, Meuwissen et al. [61] suggested updating the GS model with a large training population every cycle to increase prediction accuracy. They observed a rise in genetic gain for fertility, longevity, milk production, and other traits in cows by following this approach. Deep learning models saw the greatest improvement in forward prediction accuracy by including more training data and new environments, supporting the importance of big data for their best performance [62]. Furthermore, across location predictions were superior by using deep learning models. This could be attributed to capturing genetic, environmental, and genotype by environment interaction components by these models without explicit fitting [63]. Across location prediction can be further improved by including genotype by environment interactions or environment covariates like weather or soil parameters into the GS models to make across location and environment selections [53,64].

We observed differences in model prediction accuracies under all scenarios evaluated in this study, where machine and deep learning models performed superior to Bayesian and RRBLUP models. This difference in model performance is attributed to the different genetic architecture of each trait, dependent upon the heritability and number of QTLs controlling that trait [65]. Similar results were obtained by various other studies showing the superiority of machine learning models over conventional Bayesian models in wheat [66,67,68]. Hu et al. [69] showed that random forest performed superior to the Bayesian and RRBLUP for predicting thousand kernel weight, grain protein content, and sedimentation volume in wheat under forward prediction scenario, further strengthening our findings that machine and deep learning models should be explored for such conditions. Furthermore, we observed that highly heritable traits in this study have higher prediction accuracy than moderately heritable traits, suggesting that in addition to genetic architecture, the heritability of a trait also plays an important role in final prediction accuracy [52,70].

Machine and deep learning models performed better than all Bayesian and RRBLUP models under cross-validation, forward, and across location predictions. The higher prediction accuracy observed due to deep and machine learning models is attributed to their flexibility in deciphering complex interactions between responses and predictors to capture different trends present in the datasets compared to only additive variation in conventional GS models [71]. Deep and machine learning models explore the whole feature space during model training using different sets of neurons, activation function, and various other hyperparameters to identify the best pattern for giving the best prediction scenario compared to Bayesian models that include a pre-selected prior distribution for final predictions. Furthermore, most of the traits were predicted best by different deep and machine learning due to their respective genetic architecture of each trait. Some studies in wheat reported that all models give the same prediction accuracy irrespective of the model used while others strengthen the superiority of different models for different traits [72,73]. Ma et al. [25] and Montesinos-López et al. [26] also obtained similar results to predict quantitative traits in wheat and suggested that deep learning models should be explored due to their better prediction accuracies.

It is believed that machine and deep learning models should be used on very large training datasets, which is often not possible for end-use quality traits that are evaluated at later stages of the breeding process. However, this and other studies have shown that even small datasets can give equivalent or superior performance to the traditional parametric GS models [21,25,74]. Moreover, Bellot et al. [75] have used a training set of 100,000 individuals and showed no advantage of deep learning models over the conventional GS models. Montesinos-Lopez et al. [76] and Liu et al. [23] showed the superiority of different deep learning algorithms over conventional GS models using population sizes of 268 wheat and 4294 soybean lines. These results provide evidence that training datasets play a minor role in prediction compared to the genetic architecture of the trait. The main issue with using a small dataset for deep learning models is overfitting, resulting in the model’s failure to learn the exact pattern from the dataset [71]. Herein, we used regularization and dropout functions to remove a certain number of neurons during model training to avoid the overfitting problem [44,77].

## 5. Conclusions

We assessed the potential of machine and deep learning genomic selection models for predicting fourteen different end-use quality traits at two locations in a soft white winter wheat breeding program. Different cross-validation, forward, and across location prediction scenarios were tried for comparing different models and utilization of this approach in the breeding program. Owing to limited seed availability, time constraint, and associated cost, phenotyping for quality traits is delayed to later generations. However, the higher accuracy of prediction models observed in this study suggest that selections can be performed earlier in the breeding process. Machine and deep learning models performed better than Bayesian and RRBLUP genomic selection models and can be adopted for use in plant breeding programs, regardless of dataset sizes. Furthermore, the increase in forward prediction accuracy with the addition of more lines in the training set concluded that genomic selection models should be updated every year for the best prediction accuracy. Overall, this and previous studies showed the benefit of implementing genomic selection with machine and deep learning models for different complex traits in large scale breeding programs using collected phenotypic data from previous years.

## Figures and Tables

**Figure 1 biology-10-00689-f001:**
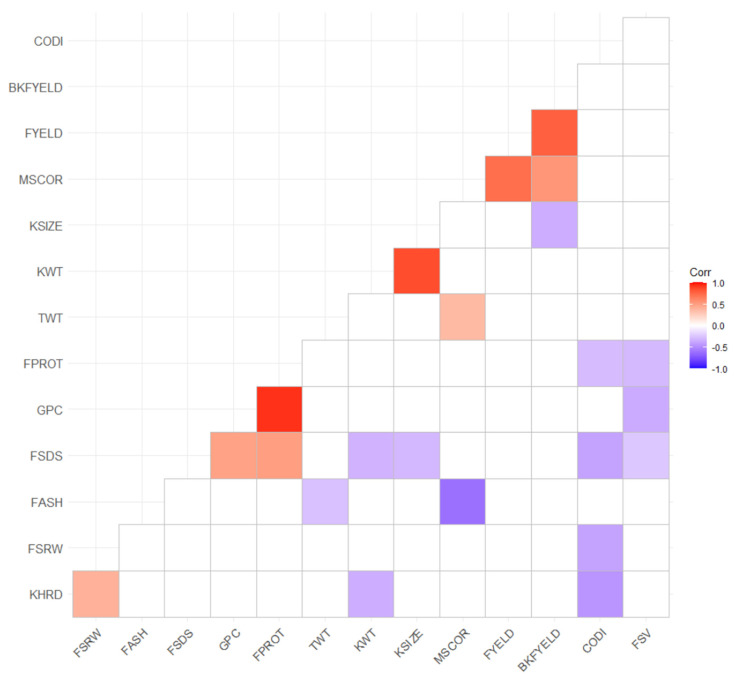
Significant phenotypic correlation between different end-use quality traits evaluated across two locations in Washington and five years using best linear unbiased predictors. All the abbreviation are previously abbreviated in the text and Table 2.

**Figure 2 biology-10-00689-f002:**
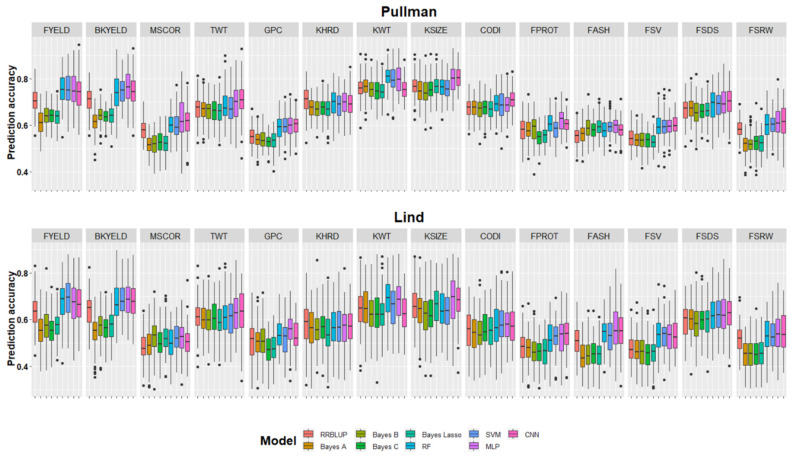
Genomic selection cross-validation prediction accuracies for fourteen end-use quality traits evaluated with nine different models. Results are provided separately for both locations and each trait is separated with facets.

**Figure 3 biology-10-00689-f003:**
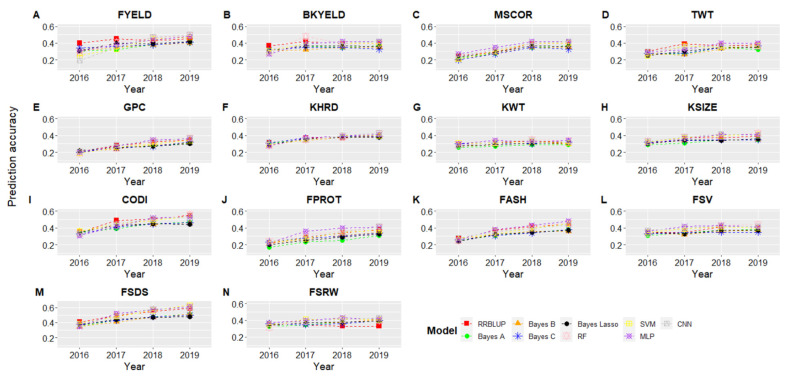
Genomic selection forward prediction accuracies for Pullman, WA, when all datasets from previous years were included to predict fourteen end-use quality traits using nine different models. The *x*-axis represents the year for which predictions were made using previous years as training set. All abbreviations are previously abbreviated in the text and Table 2.

**Figure 4 biology-10-00689-f004:**
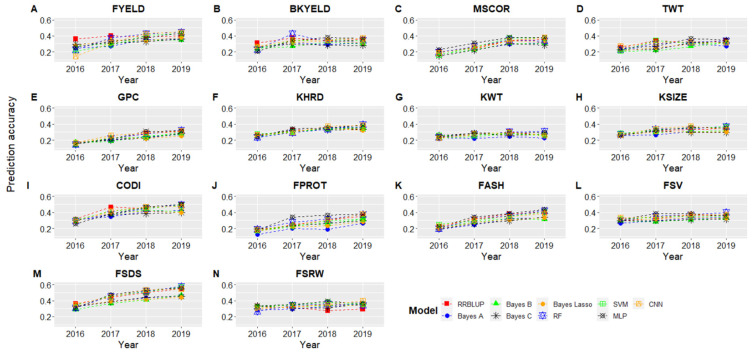
Genomic selection forward prediction accuracies for Lind, WA, when all datasets from previous years were included to predict fourteen end-use quality traits using nine different models. The *x*-axis represents the year for which predictions were made using previous years as the training set. All abbreviations are previously abbreviated in the text and Table 2.

**Figure 5 biology-10-00689-f005:**
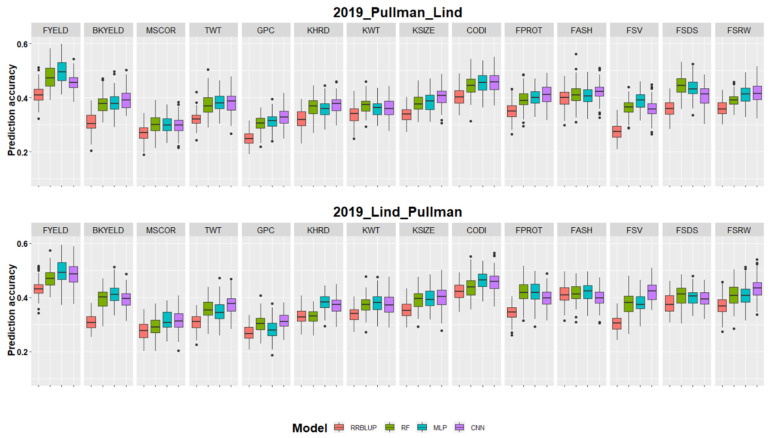
Genomic selection across environment prediction accuracies for fourteen end-use quality traits evaluated with four different models. 2019_Pullan_Lind denotes the scenario where 2019_Pullman was predicted using datasets from Lind as training set and vice versa for 2019_Lind_Pullman. Results are provided separately for both locations and each trait is separated with facets.

**Table 1 biology-10-00689-t001:** Total number of lines screened across each year at two locations in Washington and phenotyped for end-use quality traits.

Location	Year	Lines Screened for Quality
Lind	2015	122
	2016	114
	2017	115
	2018	71
	2019	106
Pullman	2015	183
	2016	128
	2017	181
	2018	137
	2019	178
Total		1335

**Table 2 biology-10-00689-t002:** Summary of the fourteen end-use quality traits evaluated for genomic selection analysis using nine different prediction models.

Trait	Abbreviation	Units	Number of Genotypes	Mean	Min	Max	S.E.	H^2^	h^2^
Milling traits									
FYELD	Flour yield	percent	666	69.9	58.0	75.8	0.09	0.91	0.75
BKYELD	Break flour yield	percent	666	48.1	33.9	56.6	0.14	0.93	0.72
MSCOR	Milling score	unitless	646	85.6	69.1	98.8	0.10	0.81	0.77
Grain characteristics									
TWT	Test weight	Kg/hL	666	61.8	54.6	65.9	0.06	0.92	0.66
GPC	Grain protein content	percent	666	10.73	7.2	14.8	0.05	0.56	0.50
KHRD	Kernel hardness	unitless	666	23.0	−10.2	52.4	0.4	0.93	0.64
KWT	Kernel weight	mg	666	39.3	26.5	54.6	0.17	0.86	0.75
KSIZE	Kernel size	mm	666	2.76	2.3	3.3	0.005	0.83	0.77
Baking parameters									
CODI	Cookie diameter	cm	622	9.2	7.8	10.0	0.008	0.89	0.82
Flour parameters									
FPROT	Flour protein	percent	666	8.93	6.3	13.0	0.04	0.57	0.46
FASH	Flour ash	percent	646	0.39	0.21	0.54	0.001	0.88	0.73
FSV	Flour swelling volume	mL/g	665	19.06	14.0	26.3	0.05	0.63	0.59
FSDS	Flour SDS sedimentation	g/mL	666	10.1	3.5	18.3	0.09	0.92	0.85
FSRW	Water solvent retention capacity	percent	666	54.18	43.4	72.6	0.09	0.85	0.77

S.E. is standard error, H^2^ is broad sense heritability, h^2^ is narrow sense heritability.

**Table 3 biology-10-00689-t003:** Genomic selection cross-validation prediction accuracies for the fourteen end-use quality traits evaluated with nine different models at two locations in Washington. The highest accuracy for each trait is bolded under different model scenarios.

Location	Trait	RRBLUP	BayesA	Bayes B	Bayes C	Bayes Lasso	RF	SVM	MLP	CNN
Pullman	FYELD	0.71 b	0.61 d	0.64 c	0.64 c	0.63 c	**0.76 a**	**0.76 a**	0.75 a	0.74 a
	BKYELD	0.70 b	0.62 d	0.64 c	0.64 cd	0.64 cd	0.75 a	0.75 a	**0.76 a**	0.75 a
	MSCOR	0.58 c	0.52 d	0.52 d	0.53 d	0.52 d	0.60 abc	0.60 bc	**0.63 a**	0.61 ab
	TWT	0.67 c	0.67 c	0.66 c	0.66 c	0.66 c	0.68 abc	0.67 bc	**0.70 ab**	**0.70 a**
	GPC	0.55 b	0.54 bc	0.54 bc	0.53 c	0.53 c	0.59 a	**0.60 a**	**0.60 a**	**0.60 a**
	KHRD	**0.71 a**	0.67 bcd	0.67 cd	0.68 bcd	0.67 d	0.70 ab	0.69 abcd	0.70 ab	0.69 abc
	KWT	0.76 b	0.77 b	0.75 b	0.75 b	0.75 b	**0.81 a**	0.80 a	0.80 a	0.75 b
	KSIZE	0.77 b	0.75 bc	0.74 c	0.75 bc	0.77 b	0.76 bc	0.76 bc	0.80 a	**0.81 a**
	CODI	0.67 bc	0.67 bc	0.67 c	0.68 bc	0.67 c	0.69 ab	0.69 abc	0.69 ab	**0.71 a**
	FPROT	0.58 c	0.58 c	0.58 bc	0.55 d	0.55 d	0.61 a	0.58 c	**0.62 a**	0.60 ab
	FASH	0.55 d	0.56 cd	**0.59 ab**	0.58 ab	**0.59 ab**	0.58 abc	**0.59 a**	**0.59 a**	**0.59 bc**
	FSV	0.55 b	0.54 b	0.53 b	0.53 b	0.53 b	0.59 a	**0.60 a**	**0.60 a**	**0.60 a**
	FSDS	0.67 de	0.67 bcde	0.66 e	0.66 e	0.67 cde	0.69 abcd	0.69 abc	**0.70 ab**	**0.70 a**
	FSRW	0.58 b	0.52 c	0.52 c	0.52 c	0.52 c	0.60 ab	0.60 ab	0.61 a	**0.62 a**
Lind	FYELD	0.64 b	0.55 c	0.58 c	0.56 c	0.58 c	0.68 a	**0.69 a**	0.67 ab	0.67 a
	BKYELD	0.63 b	0.55 c	0.57 c	0.56 c	0.57 c	0.67 a	0.68 a	**0.69 a**	**0.69 a**
	MSCOR	0.48 c	0.49 bc	**0.53 a**	0.50 b	0.52 a	0.50 b	0.52 a	0.52 a	0.50 ab
	TWT	0.61 ab	0.61 ab	0.60 b	0.61 ab	0.60 b	0.61 ab	0.61 ab	0.63 ab	**0.64 a**
	GPC	0.51 b	0.51 b	0.51 b	0.47 b	0.47 b	0.54 a	0.52 a	**0.55 a**	0.53 a
	KHRD	**0.58 a**	0.56 bc	0.56 bc	0.57 ab	0.54 c	0.56 bc	0.57 abc	0.57 abc	0.57 abc
	KWT	0.65 bc	0.65 bc	0.63 c	0.63 c	0.63 c	**0.70 a**	0.66 ab	0.69 a	0.63 bc
	KSIZE	0.66 bc	0.64 c	0.62 c	0.63 c	0.66 bc	0.64 c	0.64 c	**0.69 a**	0.68 ab
	CODI	0.56 b	0.54 b	0.54 b	0.56 b	0.55 b	0.57 ab	**0.58 ab**	**0.58 ab**	**0.58 a**
	FPROT	0.48 c	0.48 c	0.46 d	0.46 d	0.46 d	0.51 b	0.53 ab	0.53 ab	**0.54 a**
	FASH	0.51 c	0.44 d	0.44 d	0.45 d	0.44 d	0.54 ab	0.53 b	**0.56 a**	0.53 b
	FSV	0.48 b	0.47 bc	0.46 c	0.45 c	0.46 c	**0.54 a**	**0.54 a**	0.53 a	0.53 aa
	FSDS	0.59 c	0.60 c	0.59 c	0.60 bc	0.59 c	0.62 ab	**0.63 a**	**0.63 a**	0.62 ab
	FSRW	0.52 b	0.45 c	0.45 c	0.45 c	0.46 c	0.53 ab	0.53 a	**0.54 a**	**0.54 a**
Average		0.61	0.58	0.58	0.58	0.58	0.63	0.63	0.64	0.63

All the abbreviation are previously abbreviated in the text and Table 2. Models labelled with the same letter are not significantly different for each trait (*p* value = 0.05) using Tukey’s test.

**Table 4 biology-10-00689-t004:** Genomic selection across environment prediction accuracies for fourteen end-use quality traits evaluated with four different models. 2019_Pullan_Lind denotes the scenario where 2019_Pullman was predicted using datasets from Lind as the training set and vice versa for 2019_Lind_Pullan. The highest accuracy for each trait is bolded under different model scenarios.

Location	Trait	RRBLUP	RF	MLP	CNN
**2019_Pullman_Lind**	FYELD	0.41 d	0.48 b	**0.50 a**	0.46 c
	BKYELD	0.31 c	0.38 b	0.38 b	**0.40 a**
	MSCOR	0.27 b	**0.30 a**	**0.30 a**	**0.30 a**
	TWT	0.32 b	0.37 a	**0.38 a**	**0.38 a**
	GPC	0.25 c	0.30 b	0.31 b	**0.33 a**
	KHRD	0.32 c	0.37 ab	0.36 b	**0.38 a**
	KWT	0.34 b	**0.37 a**	0.36 a	0.36 a
	KSIZE	0.34 c	0.38 b	0.38 b	**0.40 a**
	CODI	0.40 c	0.45 b	**0.46 a**	**0.46 a**
	FPROT	0.35 c	0.40 b	0.40 b	**0.41 a**
	FASH	0.40 b	0.41 ab	0.41 ab	**0.42 a**
	FSV	0.27 c	0.36 b	**0.39 a**	0.36 b
	FSDS	0.36 c	**0.44 a**	0.43 a	0.41 b
	FSRW	0.36 c	0.39 b	0.41 a	**0.42 a**
**2019_Lind_Pullman**	FYELD	0.43 c	0.47 b	**0.50 a**	0.49 a
	BKYELD	0.31 b	0.40 a	**0.41 a**	0.40a
	MSCOR	0.28 b	0.29 b	**0.31 a**	**0.31 a**
	TWT	0.31 c	0.36 ab	0.35 b	**0.37 a**
	GPC	0.27 b	0.30 a	0.28 b	**0.31 a**
	KHRD	0.33 b	0.33 b	**0.38 a**	0.37 a
	KWT	0.34 b	0.37 a	**0.38 a**	0.37 a
	KSIZE	0.35 b	0.39 a	**0.40 a**	**0.40 a**
	CODI	0.42 c	0.44 b	**0.46 a**	**0.46 a**
	FPROT	0.34 c	**0.42 a**	**0.42 a**	0.40 b
	FASH	0.41 a	**0.42 a**	**0.42 a**	0.40 b
	FSV	0.30 c	0.38 b	0.38 b	**0.42 a**
	FSDS	0.38 c	**0.41 a**	0.40 b	0.40 b
	FSRW	0.37 c	0.41 b	0.41 b	**0.43 a**
**Average**		**0.34**	**0.38**	**0.39**	**0.39**

All the abbreviation are previously abbreviated in the text and Table 2. Models labelled with the same letter are not significantly different for each trait (*p* value = 0.05) using Tukey’s test.

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
