# Peer review of "Genomic Selection for End-Use Quality and Processing Traits in Soft White Winter Wheat Breeding Program with Machine and Deep Learning Models"

_biology, 2021, doi:10.3390/biology10070689_

Round 1

Reviewer 1 Report

The paper invistigated the potential of genomic selection for end-use quality and processing traits on 666 winter wheat genotypes and they compared multiple models. The study was designed well and I beleive that it can be accepted for publication after addressing major comments. I mainly concern about the significance of the differences between rrblup and ML/DL models.

1- Line 32: Need to introduce the "forward" prediction. Maybe you can reduce the first background and the conclusion parts in the abstract as they are too long and focus more on the methods and the results.

2- Introduction: While reading the introduction, I had a feeling that I'm reading a thesis not a research article. Please shorten the introduction and keep only brief descriptions for the parts that directly introduce your methods and results. For example, you do not really need to introduce the difference between soft and hard wheats.

3- Line 225: How did you fit the season within each location in your models? Did you conducted any analysis using both locations in the reference? Please add more information

4- Line 250: I think you means not all markers contribute ...

5- Line 253: we commonly call it "double exponential distribution. And please also add "respectively"

6- Figure 2: You need to calculate if the differences between rrblub and machine and deep learning models were significant. 

7- Lines 402-403: it does not seem to be always the case. Please provide a table for the results. This is very important because ML/DL usually consider gxe and non-additive variation. With the forward approach, you can show that the prediction accuracy will continue to be significantly higher even when predicting future generations.  

8- Figure 4: It does not seem that the legend is consistent with the lines. Please double check the figure.

9- Line 514: Maybe use fitted or modelled instead of programming

Author Response

1. Line 32: Need to introduce the "forward" prediction. Maybe you can reduce the first background and the conclusion parts in the abstract as they are too long and focus more on the methods and the results.

Information about forward prediction has been added in the abstract by reducing the background as suggested.

2.  Introduction: While reading the introduction, I had a feeling that I'm reading a thesis not a research article. Please shorten the introduction and keep only brief descriptions for the parts that directly introduce your methods and results. For example, you do not really need to introduce the difference between soft and hard wheats.

Introduction is cut short as suggested.

3.  Line 225: How did you fit the season within each location in your models? Did you conducted any analysis using both locations in the reference? Please add more information

Yes, analysis was conducted separately for each season within each location and cross-validation results were average to give the final prediction accuracy.

4.  Line 250: I think you means not all markers contribute ...

Change is made as suggested.

5.  Line 253: we commonly call it "double exponential distribution. And please also add  "respectively"

 These changes are made as suggested.

6.  Figure 2: You need to calculate if the differences between rrblub and machine and deep     learning models were significant. 

We didn’t report significant values as we were not promoting the use of one model over the other. Significant number in GS models can easily change depending upon the number of replications you performed. After seeing the Crossa et al., 2020, Montesinos-Lopez 2018 and various other studies which compared GS models, they do not generally report the significant difference, so we thought of not reporting that.

7.  Lines 402-403: it does not seem to be always the case. Please provide a table for the results. This is very important because ML/DL usually consider gxe and non-additive variation. With the forward approach, you can show that the prediction accuracy will continue to be significantly higher even when predicting future generations.  

As putting all this information in table in the manuscript is difficult (not to size) that’s why we decided to put this in supplementary file for reference, as you suggested.

8.  Figure 4: It does not seem that the legend is consistent with the lines. Please double check the figure.

Legends are corrected in the Figure 4 as suggested.

9.  Line 514: Maybe use fitted or modelled instead of programming

Change is made as suggested.

Reviewer 2 Report

I would like to thank the authors for this nice reading paper. This study is very timely for plant breeding and touch one of the most important aspects of breeding for nutritional traits which most of the time is considered to be secondary or comes after the selection is already done based only on yield. The paper is very well written and clear. The parts describing the GS methods and cross-validation process are clear and easy to read for non-specialists. The results are promising and will be for sure used by many breeding programs globally for better quality traits’ selection.

I have however a few comments that need reply by authors acknowledging that the nature of the data used being unbalanced.

 Material and methods:

 It will be good to have a summary of environmental connectivity (common lines between locations and years).

Statistical methods:  

The authors need to inform why the block effect is considered to be fixed?

Not clear if there is a difference between checks and evaluated lines in equation 192

Not clear if BLUEs or BLUPs were used in equation 192. I assume BLUEs as the authors are talking about adjusted means.

Did the authors use the weights in these two stages GxE analyses? Or the authors used the one stage analysis as in equation line 197?

No line effect presented in equation line 197

Results:

The heritability presented for traits in Table 1 is across env I assume. How was the heritability for single trials? This is to confirm that only trials with good heritability are use in the combined analysis.

The genomic heritability is missing.

Missing significance of correlation’s coefficients in figure 1

I am not sure table 3 and figure 2 as they almost provide the same information. Figure2 maybe better option as it includes variations in the predictions’ accuracies.

Figure 3 and 4 are difficult to read, maybe increase the graphic precision or divide each one into two.

Author Response

Material and methods:

  1. It will be good to have a summary of environmental connectivity (common lines between locations and years).

As this was a breeding population, very few lines were common between all the years except check cultivars due to continuous selections.

Statistical methods:  

  1. The authors need to inform why the block effect is considered to be fixed?

Block was considered fixed, as we want to remove that component of variation before exploring the genetic variation.

  1. Not clear if there is a difference between checks and evaluated lines in equation 192

This information is added in the section as suggested.

  1. Not clear if BLUEs or BLUPs were used in equation 192. I assume BLUEs as the authors are talking about adjusted means.

Neither BLUE nor BLUPs were used due to unbalanced data set and to avoid double shrinkage. We used adjusted means extracted using residuals from the associated model.

  1. Did the authors use the weights in these two stages GxE analyses? Or the authors used the one stage analysis as in equation line 197?

One stage analysis was conducted and reported in the method section.

  1. No line effect presented in equation line 197

This information is added in the section as suggested.

Results:

  1. The heritability presented for traits in Table 1 is across env I assume. How was the heritability for single trials? This is to confirm that only trials with good heritability are use in the combined analysis.

Heritabilities values were in the same range, as most of the traits were highly heritable, so we decided to report only one value/combined heritability in the manuscript.

  1. The genomic heritability is missing.

Information is added in Table 2 as suggested.

  1. Missing significance of correlation’s coefficients in figure 1

Significant correlations are added in the Figure 1.

  1. I am not sure table 3 and figure 2 as they almost provide the same information. Figure2 maybe better option as it includes variations in the predictions’ accuracies.

Reviewer 1 suggested providing both table and figure. So, we would like to keep both Figure 2 and Table 3 in the manuscript.

  1. Figure 3 and 4 are difficult to read, maybe increase the graphic precision or divide each one into two.

New figures are made with high resolution and updated in the manuscript.

Round 2

Reviewer 1 Report

I still feel that the author did not answer my comments. Moreover, I couldn't find table S1. Specifically, I'm concern regarding the followinng respond:

"We didn’t report significant values as we were not promoting the use of one model over the other. Significant number in GS models can easily change depending upon the number of replications you performed. After seeing the Crossa et al., 2020, Montesinos-Lopez 2018 and various other studies which compared GS models, they do not generally report the significant difference, so we thought of not reporting that."

In the summary and abstract as well as different occasion throughout the mansucript, you wrote the following:

"Deep models were superior to traditional statistical and Bayesian models under all the prediction scenarios."

So you did promot one model over the others.

Significance can change slightly if you run enough replicates and in these cases we usually run more replicates to make sure that we are reporting roboust conclusion. If a model performed much better than another, then you will get high significance whether you applied ten or 1000 replicates.

The differences between different scenarios in Crossa et al., 2019 and Montesinos-Lopez et al. 2018 were larger and clearer than your case or they were very comparable. When different models were comparable, both references declared that these cases are having comparable performances.

Author Response

Dear Reviewer,

We have done the Tukey test to show if the model's performances differ significantly, and all that information is now added to the manuscript. We forget to attach Supplementary Table 1 and is added now.

Thank you,
Karan